# Trends of Cochineal (*Dactylopius coccus*) Infestation as Affected by Armed Conflict, and Intervention Mechanisms for Sustainable Management in Tigray, Northern Ethiopia

**DOI:** 10.3390/plants14081228

**Published:** 2025-04-16

**Authors:** Haftay Gebreyesus Gebreziher, Simon Zebelo, Yohannes Gerezihier Gebremedhin, Gebremedhin Welu Teklu, Yemane Kahsay Berhe, Daniel Hagos Berhe, Araya Kahsay Gerezgiher, Araya Kiros Weldetnsae, Zinabu Hailu, Gebrekidan Tesfay Weldeslasse, Gebremariam Gebrezgabher Gebremedhin, Tsegay Kahsay Gebrekidan, Zaid Negash, Beira H. Meressa, Liberato Portillo

**Affiliations:** 1Department of Horticulture, Adigrat University, Adigrat 50, Ethiopia; yemane02@gmail.com; 2Department of Agriculture, Food and Resource Sciences, University of Maryland Eastern Shore, Princess Anne, MD 21853, USA; 3Department of Natural Resource Management, Adigrat University, Adigrat 50, Ethiopia; yohannes.gerezihier@adu.edu.et (Y.G.G.); danielhgs829@gmail.com (D.H.B.); araya.kahsay@adu.edu.et (A.K.G.); 4Deparment of Plant Sciences, Adigrat University, Adigrat 50, Ethiopia; gwtgere@gmail.com; 5Department of Geography and Environmental Studies, Adigrat University, Adigrat 50, Ethiopia; arayakiros2004@gmail.com; 6Department of Chemical Engineering, Adigrat University, Adigrat 50, Ethiopia; zinisheh@gmail.com; 7Department of Animal Sciences, Adigrat University, Adigrat 50, Ethiopia; gebrekidan.tes@adu.edu.et; 8Department of Environmental Sciences, Adigrat University, Adigrat 50, Ethiopia; gebre3g@gmail.com (G.G.G.); tsegay122008@gmail.com (T.K.G.); 9Department of Economics, Adigrat University, Adigrat 50, Ethiopia; znegashz@yahoo.com; 10Federal Research Center for Cultivated Plants, Institute for Epidemiology and Pathogen Diagnostics, Messeweg 11-12, 38104 Brunswick, Germany; beira.hailu@ju.edu.et; 11Department of Botany and Zoology, University of Guadalajara, Guadalajara 44100, Mexico; liberato.pmartinez@academicos.udg.mx

**Keywords:** cactus pear, cochineal, Tigray war, management approaches

## Abstract

The cactus pear (*Opuntia ficus-indica*) is a crucial plant in Tigray, northern Ethiopia, widely distributed in arid and semi-arid environments. It serves as a seasonal food, and is used in livestock feed, fencing, soil conservation, and environmental protection. Recently, the cactus pear populations in Tigray have been severely affected by an exotic insect, the cochineal (*Dactylopius coccus*). It damaged cactus pear populations in the region’s southern, southeastern, and eastern zones. The Tigray war that broke out in November 2020 exacerbated *D. coccus* infestation. A study was conducted in the eastern zone of Tigray to assess the impact of the armed conflict on the trends of this infestation and propose sustainable management approaches for sustainable cactus pear production in post-war Tigray. Both primary and secondary data were collected and analyzed. The findings revealed that *D. coccus* infestation significantly increased during the war and in the post-war period, compared to in the pre-war period. The number of districts involved and level of *D. coccus* infestation of cactus pear populations increased. The rapid spread was attributed to the interruption of pest management activities due to the armed conflict. To mitigate the spread and ensure sustainable cactus pear production, this study recommends different management approaches to manage *D. coccus* dissemination and sustainably produce cactus pear in the region, including pest prevention, suppression, or eradication.

## 1. Background

The cactus pear (*Opuntia ficus-indica* L. Mill), also known as the prickly pear, is believed to have originated in central and southern Mexico [1]. It belongs to the dicotyledonous plant family *Cactaceae*. This plant grows in a wide range of environmental conditions, being mainly widely distributed in arid and semi-arid environments, and remains green even during the dry season, making it especially suitable for cultivation in water-scarce areas [2,3,4].

The cactus pear provides multiple economic, social, medicinal, and environmental benefits [4]. It serves as a food source for humans, and is also used in livestock feed, income generation, live fencing, soil conservation, and environment protection [1,3,4,5,6,7,8].

The cactus pear (locally known as “Beles” or “Qulqual Bahri”) is believed to have been introduced to Ethiopia in the northeast of Tigray as early as 1848 [1,9,10]. It is widely distributed in the arid and semi-arid parts of Tigray [3]. Since its introduction to the region, interest in cactus pear production has grown significantly, due to its drought resistance, high biomass yield, high palatability, and tolerance to salinity [3]. In Tigray, the cactus pear covers approximately 379,338 ha, accounting for 7.4% of the total land area of the region [11]. It is one of the most valuable crops in Tigray, serving as a seasonal food source, mainly during the summer months (June to September), as a year-round livestock feed, as a live fence, and as a means for soil conservation and environmental protection [12]. Nearly all rural communities in the southern, southeastern, and eastern zones, and some parts of central zones, of Tigray use the cactus pear fruit as a stable food during the summer [3,8,12]. Given the severe shortage of livestock feed sources in this area, the cactus pear is utilized year-round either on its own or in combination with other feed sources [5,13]. The cactus pear has become an integral part of the culture and economy of Tigray, with it having a wide variety of uses [1,2,3,4,5,6,8]. Considering the importance of the plant to the livelihood of the community, [14] describes the cactus pear in this region as “a miracle plant, dromedary of the vegetation world, and the bank of life”.

Moreover, as the impact of climate change increases, cactus pear cultivation may assume greater agricultural importance in dry areas, since a significant part of the land is destined to become arid and semi-arid [3,4]. It is an environmentally valuable plant, particularly suited for arid and semi-arid regions, where it aids in environmental protection, soil enrichment, carbon sequestration, and climate resilience [15]. The cactus pear’s Crassulacean Acid Metabolism (CAM) photosynthesis efficiently absorbs CO_2_, making it a reliable carbon sink with high water efficiency compared to C3 and C4 pathways, which is ideal for arid environments [16]. Its drought tolerance further strengthens ecosystem pliability, offering a habitat for various species and promoting ecosystem stability [4]. Overall, the cactus pear plays a critical role in land restoration, biodiversity support, and sustainable climate adaptation.

Despite its multi-purpose livelihood benefits, the cactus pear is underutilized in Tigray [3,4,5,12]. In addition, the cactus pear industry in the region faces numerous constraints that limit its growth and potential. Some of these factors include a lack of improved agronomic practices, pests, inadequate research and extension support, low adoption of modern technologies, limited linkages among value chain actors, the absence of developed collection and marketing centers, and improper harvesting and post-harvest handling (poor packaging, sorting, and processing) [3].

The productivity of the cactus pear in the region has recently been severely affected by an invasive exotic insect, the cochineal (*Dactylopius coccus* Costa), which was deliberately introduced in 2004 for the production of carminic acid [1,3,12]. In 2004, the insect was allowed to multiply in open fields at three inoculation sites: Endayesus in Mekelle, Tsehafti in Wajarat, and Embachera in Mehoni. These sites served as experimental release points for cultivating cochineals for export purposes. Since its release, the insect has been unattended to and inadequately managed, and the project failed to meet its intended goal of exporting dried cochineals. As a result, the insect spread uncontrollably, causing extensive damage to the cactus pear population. By 2015, it had infested 16,000 ha of both wild populations and private plantations [11]. By 2020, the figure had risen to 91,000 ha [12]. The rapid spread of the pest has been facilitated by favorable climatic conditions (extended dry period), an abundance of cactus pear vegetation, a rapid reproduction rate, and the absence of natural enemies to keep its population below the required economic threshold. Moreover, the lack of a local quarantine system and ineffective pest management practices have exacerbated the situation, and it has escalated at an alarming rate. In the region’s southern, southeastern, and partly eastern zones, *D. coccus* has caused a total collapse of cactus pear production, depriving farmers of both fruits for human consumption and a source of livestock feed. Soil degradation due to erosion has become common in areas where the pest has destroyed cactus pears.

Moreover, the armed conflict (also called the Tigray War) that broke out in November 2020 affected the agricultural sector, resulting in the collapse of the whole system, including *D. coccus* management. For instance, as a result of the Tigray war, 81% of households lost their crops, 75% lost livestock, and 94% reported agricultural components having been looted or destroyed [17]. Moreover, the armed conflict interrupted ecosystem restoration and increased the risk of desertification. For instance, the woody vegetation cover declined from 17 to 12% in less than two years (2020–2022) [18]. Similarly, *D. coccus* management in the region was neglected during the war. The overall management of cactus pear protection and conservation was severely compromised as a result of the armed conflict. As the armed conflict caused huge damage to the agricultural sector of the region, it is imperative to investigate the effect of the conflict on the spread of *D. coccus* and devise coping mechanisms to contain the spread of the insect and rehabilitate cactus pear populations in the region. Therefore, this study aims to assess the trends of *D. coccus* dissemination before, during, and after the Tigray war, and identify strategies to prevent this insect pest’s spread to new areas and maintain its levels below the economic threshold.

### 1.1. Research Questions

This study investigates the following research questions:What were the trends of *D. coccus* infestation before the armed conflict?Did the armed conflict affect *D. coccus* infestation and spread in Tigray?What intervention mechanisms can be deployed to sustainably manage *D. coccus* and enhance cactus pear production in the region?

### 1.2. Objectives

The present study includes the following objectives:To assess the trend of *D. coccus* infestation in Tigray, and specifically in the eastern zone, prior to the armed conflict;To evaluate the impact of the Tigray war on the spread of *D. coccus*;To develop sustainable management options for *D. coccus* to ensure improved cactus pear production in Tigray.

## 2. Methodology

### 2.1. Description of the Study Area

Before the outbreak of the Tigray war, a buffer zone had been delineated in the eastern zone to prevent the spread of *D. coccus* to uninfested areas. The southern and southeastern zones were already fully infested before the armed conflict. Thus, this study focuses on the status of *D. coccus* spread after the armed conflict in the eastern zone of Tigray, northern Ethiopia, compared with the status of infestation prior to the conflict (Figure 1). Though the major emphasis of the study is on the eastern zone, data on the trends of *D. coccus* dissemination in the south and southeast of Tigray prior to the conflict are also considered in the study. The eastern zone encompasses 18 woredas and 168 districts (locally called “Kebeles”). The eastern zone lies between 13°33′ N latitude and 39°11′ E to 39°59′ E longitude [19]. Rainfall in the eastern zone has bimodal seasonality, whereby the long rainy season starts from the end of June to the beginning of September, and the short and sporadic rainy season extends from January to March. The eastern zone is rich in naturalized and cultivated cactus pear populations. It receives a mean annual rainfall of between 520 and 680 mm and a mean annual temperature of 16 to 20 °C [19]. The zone is one of the potential cactus pear-producing zones in Tigray. The smallholder farmers in the region utilize the cactus pear predominantly for seasonal food, livestock feed, live fences, soil conservation, and environmental protection.

### 2.2. D. coccus Life Cycle

The life cycle of *D. coccus* can vary based on different factors, like temperature and humidity. As the temperature increases, the life cycle becomes shorter, and the average length of a female cochineal insect’s life cycle is about 75 days in the hot season [12]. A study in Tigray showed that the average period from attachment to first molting (first instars), period from first to the second instar, period from second instar to maturity, and overall life cycle of the female cochineal are 19.4, 15.2, 40.4, and 75 days, respectively [12]. The mortality of the crawlers is high in the first weeks, then it decreases, and it can fall to zero in the maturity period [12]. The main way in which insects are dispersed from infested to uninfested areas is with the help of wind [4].

### 2.3. Method of Data Collection

Primary and secondary data were collected from the selected study areas. The primary data were gathered from key cactus pear-producing districts in the eastern zone of Tigray to determine the current occurrence, damage levels, and area coverage of *D. coccus*-infested cactus pear populations. This involved collecting data from 18 woredas (sub-zones) and 168 districts through direct field observations, interviews with agricultural experts from each district, participatory rural appraisal (PRA), and the use of ArcGIS for mapping. Field observation was conducted to document cochineal expansion, the status of cactus pears, and any disruptions or adaptive strategies in place. The PRA included the participation of two crop protection experts per woreda and five representative farmers from each district to provide information about their indigenous knowledge or practices employed for *D. coccus* management in their districts. ArcGIS was used to map the potential shifts in cactus pear area coverage as affected by *D. coccus* spread. The secondary data were also collected through a literature review of reports from each woreda’s agriculture bureau and agricultural production records documenting changes before, during, and after the conflict period. Combining these methods provided a well-rounded understanding of the impacts of the armed conflict, and facilitated the development of sustainable management strategies.

Finally, to delineate the buffer zone for the preventative approach, the percentage of *D. coccus* infestation in each district was determined as follows:$$\mathrm{Percent}\;D.\;coccus\;\mathrm{infestation}=\frac{D.\;coccus\;\mathrm{infested}\;\mathrm{cactus}\;\mathrm{pear}\;\mathrm{population}\;\mathrm{in}\;\mathrm{ha}}{\mathrm{Cactus}\;\mathrm{pear}\;\mathrm{coverage}\;\mathrm{in}\;\mathrm{ha}}\times 100$$

### 2.4. Data Analysis

Data on the number of *D. coccus*-affected districts, level of infestation, and coverage area of *D. coccus*-infested cactus pears, collected from each district before and after the armed conflict, were subjected to trend analysis using Microsoft Excel and ArcGIS 10.8.

## 3. Results and Discussion

### 3.1. Trends of D. coccus Dissemination in Tigray Before the Conflict

*D. coccus* was initially introduced at three sites, namely Endayesus (Mekelle zone), Tsehafti (southeastern zone), and Embachera (southern zone), in 2004 (Figure 2). From initial introduction to 2010, the spread of *D. coccus* to new districts was slow.

As depicted in Figure 3, *D. coccus* infested a few districts in the south and southeastern zones between the time of introduction and 2011. However, the level of *D. coccus* dissemination increased rapidly from 2011 onwards. From 2011 to 2014, *D. coccus* was disseminated to the majority of cactus pear-producing areas in south and southeastern woredas, except for Ofla and Sahartisamre (Figure 3, *D. coccus* infestation until 2011, *D. coccus* infestation until 2014). *D. coccus* infestation was also observed in 2014 in the Kilteawlaelo woreda of the eastern zone, but was immediately controlled, without causing economic damage to cactus pear populations (Figure 3B). The insect expanded to all cactus pear-producing areas of the south and southeastern zones of Tigray in 2016 (Figure 3C). By the time the war erupted in November 2020, the *D. coccus* infestation had fully devastated the cactus pear plants in the southern and southeastern zones of Tigray (Figure 3D).

The insect infestation in eastern-zone districts was kept minimal until 2020 by integrating cultural, mechanical, and public campaigns to clean infested fields, as well as chemical methods of pest management. The infestation level in the eastern zone of Tigray was limited to six woredas, and affected a limited number of cactus pear plants (2676.35 ha, i.e., 5.7% of the total cactus pear area coverage) until November 2020 (Figure 3D and Figure 4C, Table 1).

### 3.2. Trends of D. coccus Dissemination in the Eastern Zone of Tigray Before the Armed Conflict

The first infestation of cactus pear plants was detected in the eastern zone in 2014, in the Kilteawlaelo woreda (Figure 4A). Until 2016, the level of infestation was low, limited to a few districts in the Kilteawlaelo woreda (Figure 4B). Until November 2020, *D. coccus* was detected in only 43 districts out of 168 villages in the eastern zone of Tigray, infesting an estimated area of 2676.35 ha of the cactus pear population (Figure 4C). This accounted for 5.7% of the total cactus pear area coverage in the region (Figure 4C, Table 1) in areas near to southeastern zone. Infestation of *D. coccus* was recorded in four out of the eighteen woredas, namely Gerealta, Tsiraewemberta, Kilteawlaelo, and Hawzien. *D. coccus* was recorded in eight out of eight, ten out of twelve, eleven out of thirteen, and eight out of twenty-three **districts** in Gerealta, Tsiraewemberta, Kilteawlaelo, and Hawzien, respectively. These results show that most of the **districts in the** four woredas were affected by *D. coccus* (Table 1).

In terms of the infestation level, the highest level of infestation of cactus pears was observed in Gerealta (2342 ha), followed by Tsiraewemberta (217.75 ha), Kilteawlaelo (87.76 ha), and Hawzien (27.59 ha) (Table 1). The infested areas of cactus pear populations in these woredas account for 99% of the total infested area in the eastern zone of Tigray. As a result, a buffer zone was delineated in eastern zone districts proximate to the southeastern zone, covering parts of Gerealta, Kilteawlaelo, and Atsbiwemberta. The districts north of the buffer zone were expected to be kept free of insects through integrated management approaches, such as cultural, mechanical, and public campaigns to clean infested fields, and chemical methods of pest management [3].

### 3.3. The Effect of the Armed Conflict on the Management of D. coccus

#### 3.3.1. Neglected *D. coccus* Management Practices

From 2016 to 2020, prior to the outbreak of the war, the spread of *D. coccus* was confined to a few woredas (Figure 3 and Figure 4), and infestation levels were kept minimal through the implementation of various management practices. The management strategies focused on short-term preventative methods to limit the spread of the insect pest to uninfested areas, and a medium-to-long-term plan involving devising control methods for insect containment or eradication. The preventative methods involved integrated cultural management practices such as scouting, demarcated buffer zones between the infested and uninfested areas, raising awareness among producers and agricultural experts, mechanical methods such as cutting and burying *D. coccus*-infested cladodes or whole plants showing signs of cochineal infestation, burning and burying cochineal-infested cladodes or whole plants, and the use of botanicals and insecticides [3,20]. In addition, limitation of the movement of livestock and humans from infested to non-infested areas was partially implemented. The application and integration of various management methods reduced the spread of the pest before the war.

Scout experts in the buffer zone played a significant role in mobilizing the community, scouting, and preventing *D. coccus* from spreading [3]. As a result, from 2018 to 2020, *D. coccus* infestation in new areas was insignificant and limited to the infested areas. However, after the war erupted in November 2020, all the systems of *D. coccus* management in the region collapsed. The scout experts employed to supervise the management practices in the buffer zone lost their jobs. The pest management networks for *D. coccus* management, from the regional level to the Kebele level, collapsed. As a result, *D. coccus* rapidly disseminated to other districts beyond the buffer zone within two years of the start of the armed conflict, and even post conflict.

#### 3.3.2. Trends of *D. coccus* Infestation as Affected by the Armed Conflict

As a result of the conflict, the agricultural system of the region was devastated, and the same holds true for the *D. coccus* management practices. The armed conflict resulted in widespread *D. coccus* infestation in the cactus pear population. Insect-free cactus pear populations before the war became prone to *D. coccus* infestation. From November 2020 to October 2023, *D. coccus* infested cactus pear populations in 10 out of 18 woredas in the eastern zone of Tigray (Figure 4D and Table 1). The number of districts with *D. coccus*-infested cactus pear populations increased dramatically (Table 1, Figure 5). The total number of districts in the eastern zone with *D. coccus*-infested cactus pear populations increased from 43 (25.6%) in November 2020 to 63 (or 37.5%) in May 2023 and 92 (54.8%) in May 2024, out of 168 districts (Figure 5). *D. coccus* infestation was detected in all districts of Agulae town, Kilteawelaelo, Wukro town, Gerealta, and Hawzien town (Figure 5) during and following the conflict. By September 2024, *D. coccus* was detected in Gantaafeshum, Bizet, Sibuhasaesie, and Gulemekda, with no or little impact on cactus pear populations. On the other hand, Irob, Adigrat, Freweyni town, Edagahamus town, and Atsbi town had not experienced any *D. coccus* occurrence at this point.

In terms of coverage area, a total of 2676.35 ha of cactus pear populations were infested by *D. coccus* until November 2020. The level of infestation rose to 3644.79 ha (26.57% increase) in May 2023 and 5239.47 ha (48.92% increase) in May 2024 (Table 1). However, the percentage of *D. coccus* infestation varied among the woredas (Table 1, Figure 5 and Figure 6). Until May 2024, among the woredas that had been already affected by cochineals before the war, the highest rate of increase in cochineal infestation of cactus pear populations, out of the total coverage of cactus pear infestation, was observed in Kilteawlaelo (87.67%), followed by Tsiraewemberta (30.65%) (Figure 6). During the war and in the post-war period, the insect was disseminated to other woredas in the eastern zone, including Saesietsaedaemba, Sibuhasaesie, Gantaafeshum, and Bizet (Figure 6). The cactus pear populations in three towns, namely Hawzien, Wukro, and Agulae, were also infested by *D. coccus*. As depicted in Figure 6, the level of *D. coccus* infestation and the number of districts affected increased in September 2024 compared to November 2020. The alarming dissemination rate of *D. coccus* during and after the conflict is attributed to the collapse of management approaches. There is poor level of intervention for *D. coccus* management, which is threatening cactus pear production in the remaining insect-free districts of the eastern zone.

## 4. Existing Challenges in the Management of *D. coccus*

*D. coccus* is spreading to new cactus pear population districts because of existing challenges in dealing with the problem. A major challenge arises from the lack of scout deployment and public campaigns to halt the spread of the insect pest to new districts. There has been little intervention to prevent the spread or control of *D. coccus* since the eruption of the armed conflict and even after the Pretoria Agreement (a peace treaty between the government of Ethiopia and the Tigray People’s Liberation Front (TPLF) that was signed on 2 November 2022). Another challenge arises from the fast nature of dissemination of the insect pest. *D. coccus* is mainly disseminated by wind and phoresis, and thus its infestation rate is high during the dry and windy season. If preventative methods are not applied, the infestation rate is expected to rise, leading to devastating damage to the remaining cactus pear population in the region. There is a lack of integrated approaches in management methods, human resources, and experts to provide a truly effective solution to the problem. The lack of a buffer zone to prevent *D. coccus* dissemination is another challenge that favors fast dissemination of *D. coccus* to the uninfested districts if intervention is not applied as quickly as possible. Poor research and development in *D. coccus* management, cactus pear rehabilitation, and planting techniques are also contributing factors. Another challenge is the unmanageable planting nature of cactus pear populations. Wild and cultivated populations characterized by overgrowth and lack of spacing among cactus pear plants make it challenging to manage *D. coccus* through the application of pest management practices such as pesticides. Moreover, there is no clarity on whether or not harvesting and commercializing *D. coccus* for carminic acid production should be considered as a means of controlling dissemination to new districts. The most critical point here is reaching clarity among leaders, experts, institutes, and the community through an in-depth analysis of whether legalizing *D. coccus* multiplication and harvesting is a blessing or a curse. An expert-driven decision on whether harvesting *D. coccus* should be considered as part of cochineal management to create a better environment is important.

## 5. The Way Forward for Integrated *D. coccus* Management to Restore and Sustainably Produce Cactus Pears in Tigray

*D. coccus* has become one of the most invasive insect pests, causing damage to vast areas of neutralized and cultivated cactus pear populations in Tigray. Dissemination of the insect occurs at an alarming rate that demands action and the development of integrated *D. coccus* management. Given the nature of the cactus pear population in the region and the fast dissemination of the insect pest, as depicted in this study, it is paramount to implement approaches for integrated *D. coccus* management and sustainable production of cactus pears in Tigray. Proposed management approaches are intentionally incorporated in this study, in order to contribute to informed decision-making regarding *D. coccus* management and sustainable cactus pear production in the region.

### 5.1. Preventative Approach

To save the remaining cactus pear population in Tigray, the first step in *D. coccus* management is expected to focus on preventative methods to halt the dissemination of the insect pest to uninfested districts. Before the armed conflict, preventative methods involved delineating a buffer zone, scouting, and burning, cutting, and burying cladodes with signs of immature cochineals (nymphs) [3]. In addition, quarantining, ensuring hygiene in fruit packing material, and limiting movement of livestock and humans from infested to non-infested districts have been part of preventative methods. The preventative approach can be effective if the smallholder farmers who own cactus pear populations are involved and technically and financially supported, public campaigns to clean infested fields are arranged, and awareness is raised. Above all, delineating a buffer zone could be a priority to halt the overwhelming dissemination of the insect pest to other cactus pear-producing districts in the zone. While delimiting the buffer zone, it is paramount to consider the infestation levels of districts. In terms of the coverage area of *D. coccus*-infested cactus pear populations, some woredas, such as Saesietsaedaemba, Sibuhasaesie, and Gantafeshum, show low levels of infestation, while woredas such as Kilteawlaelo, Atsbiwemberta, Hawzien, and Tsiraewonberta show high levels of infestation. The *D. coccus* in the woredas with low infestation levels can be controlled by applying cultural, mechanical, and chemical methods. With the assumption that these woredas are expected to be free of cochineals or have low cochineal infestation levels below the economic threshold, a 5 km buffer zone can be demarcated between infested districts and non- or less-infested districts in the eastern zone of Tigray, as proposed in Figure 7. Based on this, towards the north of the buffer zone, integrated preventative approaches can be applied to halt the dissemination of *D. coccus* to other districts, whereas to the south of the buffer zone, suppression strategies must be developed for the already-infested districts based on the level of infestation, in order to reduce the pest population to a level the below economic threshold.

### 5.2. Pest Suppression Approach

Apart from the preventative approach, in districts where *D. coccus* is causing economic loss, it is recommended to apply management mechanisms that reduce the *D. coccus* population to a level below the economic threshold, and rehabilitating the cactus pear population in the affected districts of the region is vital. The management approaches are expected to integrate various control methods of *D. coccus*. The following management methods in a single or integrated approach are recommended to reduce or control *D. coccus*.

Awareness creation and public campaigns: There is a poor level of understanding and awareness about the biology of *D. coccus* and its management methods at different leadership levels and in the community in Tigray. The same holds for the economic opportunities and drawbacks of the introduction of *D. coccus* to the region. Therefore, raising awareness and providing training on the threats and opportunities of cochineals and their impact on cactus pears is crucial. Without public campaigns and coordinated efforts of the smallholder farmers who own the cactus pears, experts, community leaders, local and international NGOs, and the government, it is difficult to halt the fast dissemination of the insect. As part of *D. coccus* management approaches, it is imperative to mobilize smallholder farmers and collaborate with stakeholders at different levels. Before the war, awareness creation and public campaigns had been part of the integrated *D. coccus* management in the region [3].

Mechanical methods: Removing the *D. coccus*-infested cactus pear population using different techniques can be considered as one control method. This may involve cutting, burning, and burying infested cladodes or whole plants, based on the level of infestation. Reports show that mechanical methods were employed to prevent the dissemination of *D. coccus* in Tigray before the armed conflict [1,3,21].

Chemical control: A plethora of findings show that insecticides and insecticidal soaps are effective in killing various species of cochineal insects at different life stages [7,22,23,24]. However, the absence of proper spacing between and among the cactus pear population in Tigray is a challenge for applying chemical insecticides. If the cactus pear population is managed to ensure appropriate spacing and access, chemical insecticides can be effective in controlling *D. coccus*.

Harvesting and commercializing D. coccus for carminic acid production: After cochineals were introduced to Tigray in 2004, they were not harvested in time for commercial use [4]. Based on plenty of experience in utilizing both cacti and cochineals, with due consideration, it is evident that protected management is required for *D. coccus* harvesting and multiplication. Reports show that if the cochineal insects had been harvested, the amount of dye produced would have increased from 3025 tons in 2011 to 60,253 tons in 2018. From this total dye production, $24,012,200 would have been obtained from carminic acid [4]. Because of this, the introduction of cochineals to Tigray is considered a missed opportunity in a region with such a huge cactus population [4,6]. Therefore, commercializing cochineals for the production of carminic acid should not be ignored as an option for inclusion in *D. coccus* management and utilization approaches. It is time to move on from these challenges and wasted fortunes, and consider the multitude of opportunities that the region can take advantage of to gain economically by exploiting both cactus pears and *D. coccus*. The region should promote this issue as a point of discussion among different actors in the sector, conduct an in-depth analysis of management strategies, and change perceptions and raise awareness in the community about *D. coccus* commercialization and the possibilities of exploiting cactus pears and these insects to gain economic benefits. Thus, the region is expected to consider exploiting *D. coccus* for economic purposes in a scientifically proven and protected manner, as an option to reduce the pest population in the already-devastated districts of the region. The potential cactus-producing zones can be clustered according to the availability of the insects. One option is allowing the harvesting of *D. coccus* in zones with high infestation levels, while preventing the dissemination of the insect to uninfested districts, and incentivizing farmers in uninfested districts to modernize their cactus pear production and create market linkage. Another alternative is allowing investment in *D. coccus* multiplication in protected structures, such as greenhouses, while preventing open-field multiplication. Farmers in uninfested districts will be linked to firms by allowing them to supply cladodes for *D. coccus* multiplication in protected structures. Considering these and other options is paramount to make an informed decision on whether or not to commercialize cochineals for carminic acid production, and whether to consider them a blessing or a curse.

Selection of resistant cactus pear varieties: Searching for local varieties or introducing new cactus pear varieties that are resistant to *D. coccus* is also a management option to consider in order to maintain the pest population at a level below the economic threshold. The cactus species *Opuntia stricta* and *O. robusta* showed high resistance to *D. coccus* in a study conducted under laboratory and field conditions [25]. The cactus pear species in Tigray is *O. ficus-indica*, characterized by varied population types. The different population types (local cultivars) in Tigray have yet to be fully tested for their resistance. In other countries, certain cultivars of *O. ficus-indica* have been found to be resistant to *D. coccus*. For instance, [3] found that *O. ficus-indica* cv. ‘Rojo Pelon’ was resistant to *D. coccus*. In another study in Tigray, *O. robusta* and *O. stricta* were found to be resistant to *D. coccus* [12]. Therefore, introducing different resistant species or cultivars can be considered as a means to suppress *D. coccus* in Tigray.

Botanicals: Plant extracts from plants possessing insecticidal chemicals can be considered an option for *D. coccus* containment below an economic threshold. There have some trials investigating the efficacy of botanicals in increasing the mortality of cochineals. For instance, extracts of leaves of tree tobacco (*Nicotiana glauca*) caused high mortality of female *D. coccus* [20]. Another study also showed that extracts from *Solanum linnaenum* and *Nerium oleander* caused high mortality of *D. coccus* [8]. Considering these promising findings, botanicals could be an integral part of integrated *D. coccus* management.

Biological control: Searching for any natural enemies in Tigray that might be adapted to *D. coccus* is imperative. Alternatively, it is important to consider introducing exotic natural enemies, like beetles, including *Chilocorus cacti*, *Hyperaspis trifurcate*, and lepidopteran species, like *Laetilia cocci*. Reports show that *C. cacti* controls different cochineal species [26]. Research findings also show that *H. trifurcate* effectively controls cochineals [27]. Therefore, based on this evidence, considering biological control as a means of *D. coccus* control through either conventional, augmentative, or conservational methods is important.

Proper agronomic practice to grow cactus pears: To successfully implement the above control methods, in terms of appropriate agronomic practices in growing cactus pears, emphasis should be placed on establishing the new cactus populations in Tigray. The existing population in Tigray lacks proper spacing and grows in hard-to-reach places, such as at the top of mountains and on steep slopes, and this contributes to the spread of *D. coccus* in Tigray. A spacing of one to two meters between plants and two to three meters between rows is recommended for growing cacti for fruit and vegetable (cladode) production [28]. Proper spacing ensures adequate airflow, prevents disease and pest spread, reduces interspecific competition among cactus plants, and increases yield. Moreover, adequate spacing facilitates pest management by enhancing the application of pesticides and botanicals, releasing biological controls, and monitoring the effectiveness of the applied control methods.

### 5.3. Eradication Approach

Eradication is the application of phytosanitary methods to eliminate a pest from an area or geographic region [29]. It is implemented after a careful assessment of anticipated economic, ecological, and social consequences. The eradication approach is the most challenging method to employ, due to its substantial resource requirements and difficulty to implement. However, there are few success stories of pest eradication. To give a few examples among many, Myers [30] highlights the eradication of screwworms (*Cochliomyia hominivorax*) through an sterile insect release (SIR) program (1958–1960) in Florida, and cattle ticks (*Boophilus annulatus*) by exclusion, disinfection, and cattle quarantine in the USA, as successful eradication programs. In contrast, effort to implement Gypsy moth and medfly eradication programs were recorded as unsuccessful. It is understandable that it is difficult to achieve success with the eradication approach as a stand-alone technique, as it demands a multi-year support program. The Sterile Insect Technique (SIT), conventional biological control, host resistance, and other techniques can be considered as options to successfully eradicate the *D. coccus* pest. However, the eradication of cochineals from Tigray could be very difficult, given the resource limitations and high commitment demands. In this regard, maintaining the level of cochineals below the economic threshold by employing the preventative and suppressive management approach or commercializing cochineals for carminic acid production could be alternative options to the eradication approach.

## 6. Conclusions

Regardless of the immense number of livelihood functions it serves, the cactus pear is an underutilized plant in the Tigray region. It has been devastated by *D. coccus* in the last two decades. *D. coccus* completely damaged the cactus pear populations in the south and southeastern zones and partially disseminated to the eastern zone of Tigray before the armed conflict erupted in November 2020. Prior to the conflict, the spread of *D. coccus* to new districts in the eastern zone of Tigray had been effectively slowed through an integrated management approach and the establishment of a buffer zone between infested and uninfested areas. This study reveals that the conflict led to the collapse of these management practices, resulting in a significant increase in pest dissemination. The rapid spread of the insect resulted in damage to a significant coverage area of cactus pears, and disseminated to new districts with a high cactus pear population. Considering the fast-spreading nature of the insect, it is vital to apply integrated *D. coccus* management approaches, including pest prevention, suppression, and eradication, which could contribute to sustainable cactus pear production, while changing the challenges posed by the insect into an economic opportunity by developing a strategy for protected *D. coccus* harvesting and commercialization. To restore and sustainably manage cactus pear populations, it is recommended to develop a strategy that promotes the exploitation of both the multi-purpose cactus pear and the *D. coccus* species, which is known for its utility in carminic acid production, by introducing protected and sustainable production approaches. In addition, further study on the impacts of climate change on the rapid invasion of *D. coccus* is crucial in order to forecast how it will be affected by the ever-increasing impact of global warming.

## Figures and Tables

**Figure 1 plants-14-01228-f001:**
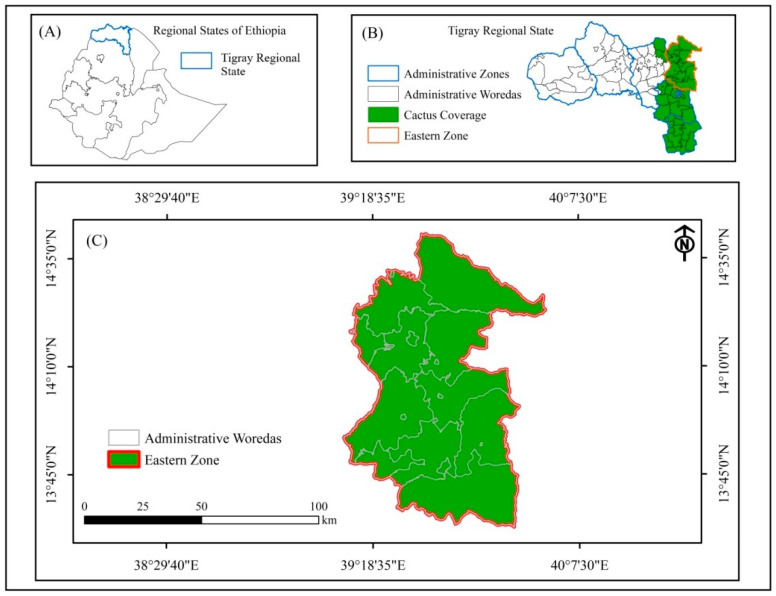
Study area: (**A**) Ethiopia’s regional states, with blue outline indicating Tigray Regional State; (**B**) Tigray Regional State; and (**C**) eastern zone.

**Figure 2 plants-14-01228-f002:**
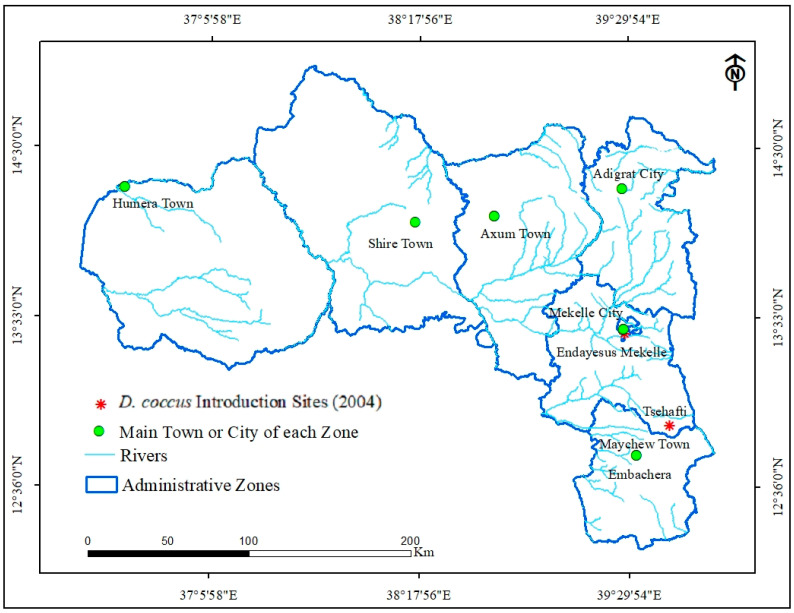
*D. coccus* introduction sites in 2004, namely Endayesus Mekelle in Mekelle city, Tsehafti in southeastern zone, and Embachera in southern zone of Tigray.

**Figure 3 plants-14-01228-f003:**
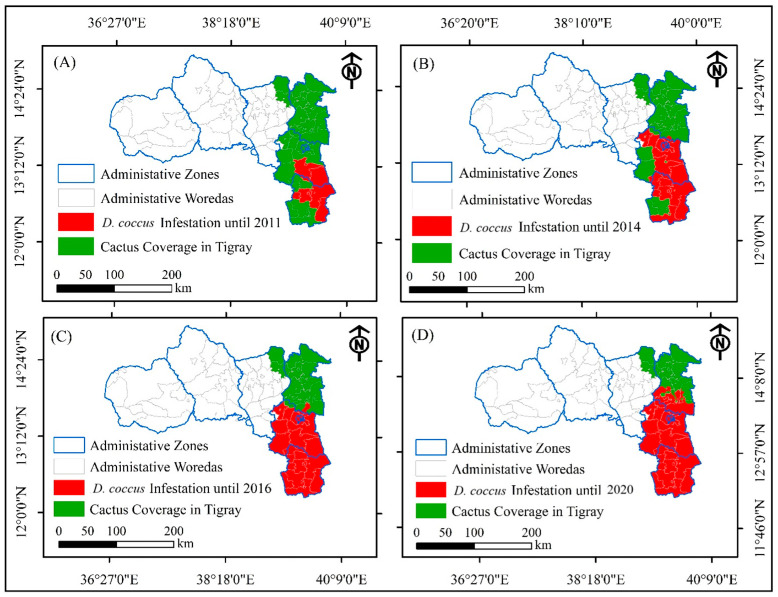
Pre-war *D. coccus* spread in cactus pear-producing areas in Tigray. *D. coccus* infestation until 2011 (**A**); *D. coccus* infestation until 2014 (**B**); *D. coccus* infestation until 2016 (**C**); *D. coccus* infestation until 2020 (**D**).

**Figure 4 plants-14-01228-f004:**
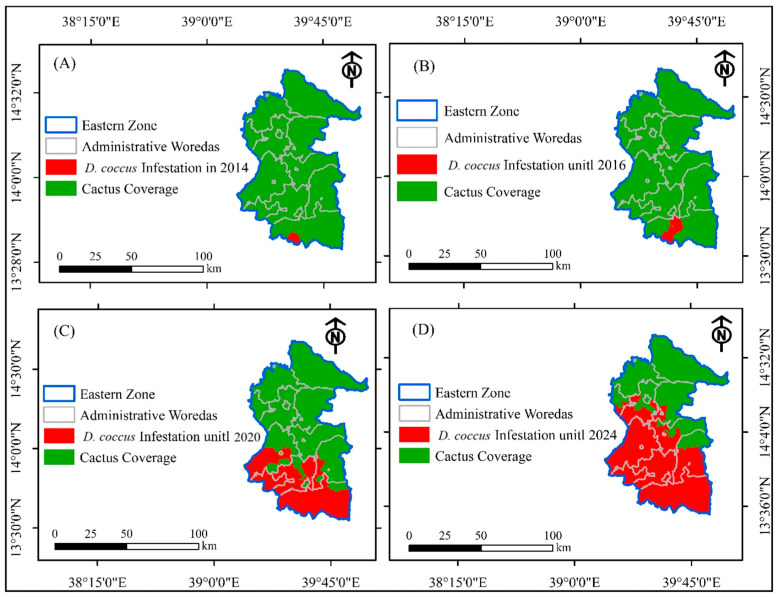
*D. coccus* expansion in the eastern zone of Tigray. (**A**) *D. coccus* infestation reported in 2014, (**B**) *D. coccus* infestation reported in 2016, (**C**) *D. coccus* infestation reported in 2020, and (**D**) *D. coccus* infestation reported in 2024.

**Figure 5 plants-14-01228-f005:**
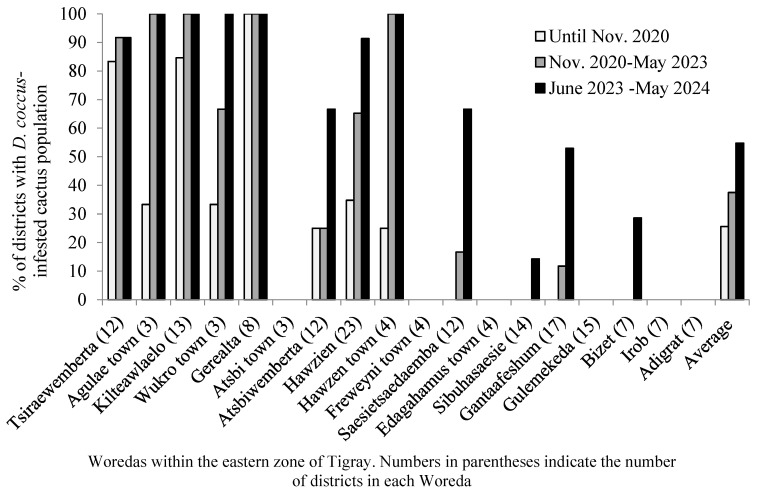
The percentage of districts with *D. coccus*-infested cactus pear populations in the eastern zone of Tigray before the war (until November 2020) and during and after the war (November 2020–May 2024).

**Figure 6 plants-14-01228-f006:**
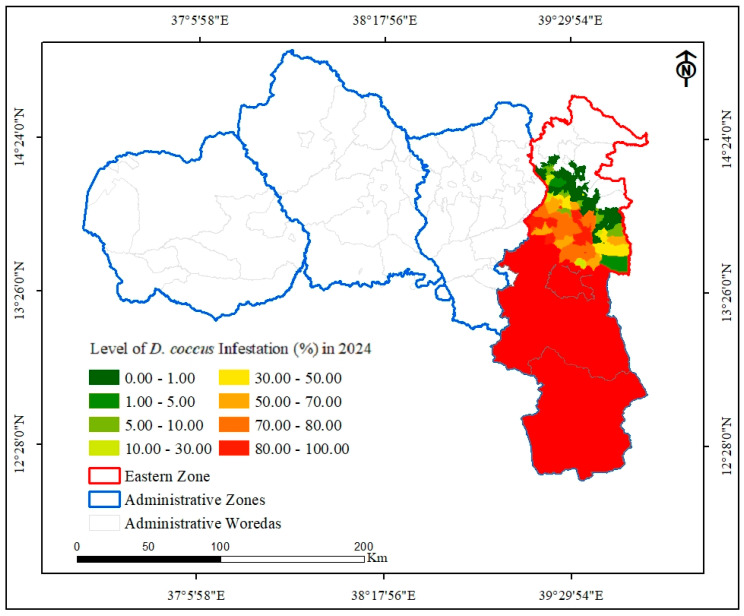
A heat map showing the level of *D. coccus* infestation in the eastern zone in May 2024.

**Figure 7 plants-14-01228-f007:**
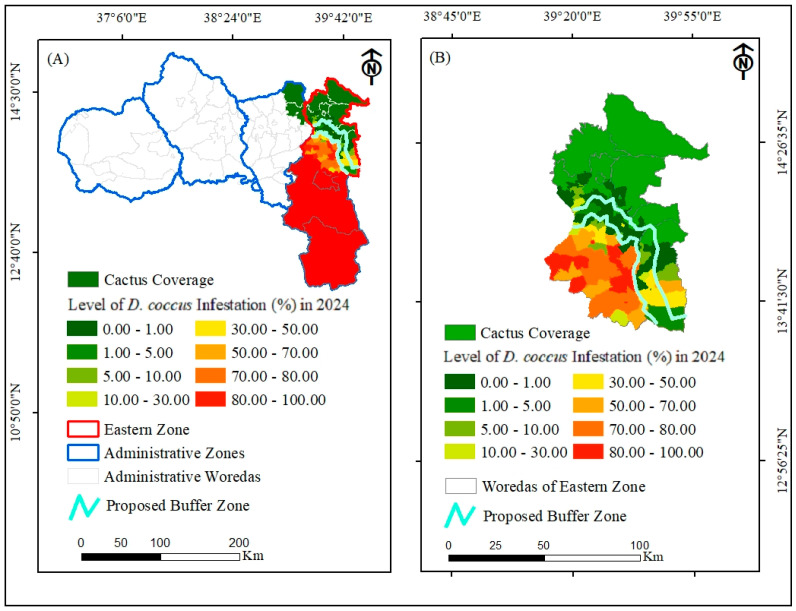
Proposed buffer zone to prevent spread of *D. coccus*. (**A**) Map of Tigray region with buffer zone delineation; (**B**) Map of eastern zone of Tigray with buffer zone delineation.

**Table 1 plants-14-01228-t001:** Trends of *D. coccus* spread in terms of the number of districts and the areas (hectares) infested in the eastern zone of Tigray (November 2020–May 2024).

Name of Woreda	Number of Districts	Cactus Pear Area Coverage (ha)	*D. coccus*-Affected Districts			*D. coccus*-Infested Cactus Pear Coverage (ha)		
			Before the War (Until November 2020)	Until May 2023	Until May 2024	Before the War (Until November 2020)	Nov 2020–May 2023	May 2023–May 2024
Tsiraewemberta	12	1353.10	10	11	11	215.75	362.00	414.75
Agulae town	3	6.50	1	3	3	1.00	6.50	6.50
Kilteawlaelo	13	2060.99	11	13	13	87.76	794.61	1806.60
Wukro town	3	4.67	1	2	3	1.50	4.67	4.67
Gerealta	8	2342.00	8	8	8	2342.00	2342.00	2342.00
Atsbi town	3	3.00	0	0	0	0.00	0.00	0.00
Atsbiwemberta	12	2549.20	3	3	8	2.25	1.75	22.14
Hawzien	23	3580.25	8	15	21	27.59	131.26	607.63
Hawzen town	4	6.50	1	4	4	1.47	6.50	6.50
Freweyni town	4	7.00	0	0	0	0.00	0.00	0.00
Saesietsaedaemba	12	4109.25	0	2	8	0.00	2.00	27.59
Edagahamus town	4	5.00	0	0	0	0.00	0.00	0.00
Sibuhasaesie	14	8321.00	0	0	2	0.00	0.00	0.80
Gantaafeshum	17	4151.50	0	2	9	0.00	0.00	5.59
Gulemekeda	15	4173.25	0	0	0	0.00	0.00	0.00
Bizet	7	664.63	0	0	2	0.00	0.00	1.20
Irob	7	14,953.50	0	0	0	0.00	0.00	0.00
Adigrat	7	75.00	0	0	0	0.00	0.00	0.00
Total	168	48,366.34	43	63	92	2676.35	3644.79	5239.47

## Data Availability

The datasets generated during and/or analyzed during the current study are available from the corresponding author on reasonable request.

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
