# Peer review of "Trends of Cochineal (Dactylopius coccus) Infestation as Affected by Armed Conflict, and Intervention Mechanisms for Sustainable Management in Tigray, Northern Ethiopia"

_plants, 2025, doi:10.3390/plants14081228_

Round 1
Reviewer 1 Report
Comments and Suggestions for Authors
Congratulations to the author for this good article on pest control and D. coccus infestation. This article discusses in detail the pest management and historical aspects of the occurrence, distribution and infestation of Dactylopius coccus before and after the war in the main pear cactus growing area in the Eastern Zone of Tigray. The team collected sufficient data including primary and secondary data focusing on 18 subzones and 168 districts and processed the trend of infestation, spread and development of buffer control zones on a map using ArcGIS. After reading this article, I understand very well what the author is trying to convey about the spread and infestation of the pest since its introduction in 2004 and a sustainable approach to control the infestation of D. coccus on cactus farms in Tigray. There are only minor corrections for the author. Further information can be found in the attached file.

Reviewer 2 Report
Comments and Suggestions for Authors
The Manuscript "Trends of Cochineal (Dactylopius coccus) Infestation as Affected by Armed Conflict and Intervention Mechanisms for Sustainable Management in Tigray, Northern Ethiopia" provides a comprehensive study on the impact of armed conflict on the spread of cochineal (Dactylopius coccus) infestation in Tigray, northern Ethiopia, and proposes sustainable management strategies. The research is timely and relevant, given the significant economic and ecological importance of cactus pear in the region and the recent challenges posed by the pest and conflict.
- The study innovatively links the impact of armed conflict on pest management, an area that is often overlooked in ecological research. This provides a novel perspective on how socio-political instability can exacerbate biological invasions.The background information on cactus pear and cochineal infestation is detailed in the introduction section, but it could benefit from a more structured approach to highlight the progression of the problem. For example, the historical introduction of cochineal and its initial impact could be better connected to the current situation, emphasizing the urgency and relevance of the study.
- The proposed integrated management approaches, including both preventative and suppressive methods, offer a comprehensive framework for addressing the cochineal infestation problem. The consideration of commercializing cochineal for carminic acid production as a potential management strategy is particularly innovative.
Suggestions for Improvement "Despite existing studies on cochineal infestation in Tigray, there is a significant gap in understanding how the recent armed conflict has exacerbated the spread of D. coccus and impacted cactus pear populations. This study aims to fill this gap by assessing the trends of D. coccus infestation before, during, and after the Tigray war and identifying sustainable management approaches."
- Revise the objectives to make them more specific and directly linked to the research gap. Add a section that outlines the research questions or hypotheses.
- The figure 1 lacks clear labels and legends to distinguish between different zones and regions. It is difficult to immediately understand the boundaries and the specific study area within the larger context of Ethiopia. Add clear legends and labels to differentiate between Ethiopia’s regional states, Tigray Regional State, and the eastern zone. Use distinct colors or shading to highlight the study area and ensure it is easily distinguishable.
- Figure 2: Enhance the map with more detailed geographical features (e.g., rivers, roads, major towns).
- Figures 1 and 2 could be considered combined.
- The figure 5 lacks clear labels for the x-axis and y-axis. The legend is not detailed enough to explain the different periods represented by the bars.
- While the manuscript is of high quality, there are a few areas that could be improved. The authors could provide more detailed information on the specific challenges faced during data collection and how these were addressed. Additionally, a discussion on the potential long-term ecological impacts of cochineal infestation on the broader ecosystem in Tigray would strengthen the manuscript.
- Results and Discussion can be separated.
The discussion section does not clearly link the results to the objectives outlined in the introduction. While the trends of cochineal infestation and the impact of the armed conflict are discussed, there is no explicit connection to the stated objectives, making it difficult for readers to understand how the results address the research questions. the increase in cochineal infestation during and after the conflict is mentioned, but there is no exploration of the underlying ecological or socio-economic factors that may have contributed to this trend. The discussion section does not address the limitations of the study. Limitations should be acknowledged, and their potential impact on the results should be discussed.
Overall, the manuscript is well-researched, well-written, and provides valuable insights into the complex issue of cochineal infestation in Tigray. It meets the journal's requirements and is likely to be of interest to researchers and practitioners in the fields of ecology, agriculture, and sustainable development. With minor revisions to address the points mentioned above, this manuscript has the potential to make a significant contribution to the literature.
